# Selective capture of carbon dioxide from hydrocarbons using a metal-organic framework

Omid T. Qazvini[1,2], Ravichandar Babarao [3,4] & Shane G. Telfer [1✉]

Efficient and sustainable methods for carbon dioxide capture are highly sought after. Mature technologies involve chemical reactions that absorb $CO_2$, but they have many drawbacks. Energy-efficient alternatives may be realised by porous physisorbents with void spaces that are complementary in size and electrostatic potential to molecular $CO_2$. Here, we present a robust, recyclable and inexpensive adsorbent termed MUF-16. This metal-organic framework captures $CO_2$ with a high affinity in its one-dimensional channels, as determined by adsorption isotherms, X-ray crystallography and density-functional theory calculations. Its low affinity for other competing gases delivers high selectivity for the adsorption of $CO_2$ over methane, acetylene, ethylene, ethane, propylene and propane. For equimolar mixtures of $CO_2/CH_4$ and $CO_2/C_2H_2$, the selectivity is 6690 and 510, respectively. Breakthrough gas separations under dynamic conditions benefit from short time lags in the elution of the weakly-adsorbed component to deliver high-purity hydrocarbon products, including pure methane and acetylene.

[1] MacDiarmid Institute for Advanced Materials and Nanotechnology, School of Fundamental Sciences, Massey University, Palmerston North, New Zealand. [2] Department of Chemical Engineering and Analytical Science, The University of Manchester, Oxford Road, Manchester M13 9PL, UK. [3] School of Science, RMIT University, Melbourne, VIC 3001, Australia. [4] Commonwealth Scientific and Industrial Research Organisation (CSIRO) Manufacturing, Clayton, VIC 3169, Australia. ✉email: s.telfer@massey.ac.nz

Chemical separation processes consume vast quantities of energy[1]. Economical and practical pathways to alleviating this burden are required. This is especially relevant to the capture of $CO_2$, which is a common impurity in crude gas streams. $CO_2$ removal is integral to upgrading natural gas and biogas, for example, and to the purification of valuable hydrocarbons prior to polymerisation or chemical derivatization[2]. These processes are separations that rely on discrimination between $CO_2$ and other gases. One established technology is to trap the $CO_2$ by a chemical reaction with an absorbent. This typically involves chemisorption to an amine in aqueous solution[3,4]. Chemisorption incurs multiple drawbacks, however, including a high energy penalty during regeneration, amine losses due to degradation and evaporation, and the corrosion of hardware and pipelines[5]. Other conventional separation methods involve solvent extraction or cryogenic distillation, which are burdened with a high energy penalty and large amount of solvent waste.

The physisorption of $CO_2$ in nanoporous materials is an attractive alternative[6,7]. Physisorption is governed by weak, noncovalent bonding interactions in pores that are structured on the molecular scale[8]. Ideally, they lower the energy requirements for regeneration since driving off the trapped $CO_2$ simply involves breaking interactions that are inherently weak. Effective physisorbents combine rapid guest diffusion, recyclability and long-term stability with selectivity for $CO_2$ over competing gases at relevant concentrations[9]. Thus, they may offer a sustainable solution to $CO_2$ capture. In this context, metal-organic frameworks (MOFs) have risen to prominence[10–14]. MOF materials are built up from metal ions and organic ligands, and their pore shape, size and chemical environment can be systematically designed[15,16]. In turn, this allows interactions between framework hosts and molecular guests to be tailored. In the search of effective MOF physisorbents, simply searching for materials with ever-higher levels of $CO_2$ uptake per se may not deliver adsorbents that are adept at gas separations since the adsorption of non-$CO_2$ components may also increase. Instead, significant advances will emerge by suppressing the uptake of these competing gases[17,18], developing scalable synthetic protocols, mitigating the impact of common impurities such as water vapour and oxygen, and developing low energy pathways to adsorbent recycling.

The removal of $CO_2$ from hydrocarbons is an important process[2]. While natural gas and biogas are primarily composed of methane (at high pressure and low pressure, respectively), contamination by $CO_2$ can prevent optimal heat release from gas combustion, and cause pipeline corrosion and dry ice formation[19]. MOFs, however, offer a means of reducing the $CO_2$ concentration in the presence of dominant quantities of methane[10,20,21]. Acetylene ($C_2H_2$) is an essential feedstock for the industrial production of commodity materials[22,23]. When acetylene is generated, however, it typically coexists with $CO_2$ impurities[24]. The separation of $C_2H_2$ and $CO_2$ is challenging due to their similar physical properties (Supplementary Table 4). MOF physisorbents offer a potential solution but most show an affinity toward $C_2H_2$ rather than $CO_2$[11]. The selective adsorption of the $CO_2$ component has seldom been reported despite its operational simplicity in process design and the promise of energy efficiency. Conversely, gas purification using hydrocarbon-selective MOFs requires additional stages if the eluent is contaminated by adsorbed $CO_2$ during the desorption step[25]. Despite recent advances in MOF chemistry, challenges remain in producing framework adsorbents that combine good separation capabilities with wider performance characteristics such as scalability, recyclability and easy low-energy regeneration. MOF adsorbents that may be applied to methane purification and that preferentially adsorb $CO_2$ from other hydrocarbons are in particular demand.

In this work, we present a MOF, termed MUF-16 (MUF = Massey University Framework) that exhibits inverse selectivity: the adsorption of carbon dioxide in preference to hydrocarbon guests. The carbon dioxide is efficiently sequestered by hydrogen bonding and a range of other favourable noncovalent interactions. This underpins high selectivities for the separation a range of gas mixtures that are relevant to natural gas and industrial feedstocks. Being economical to produce on scale, stable and recyclable, MUF-16 has many of the qualities of an attractive adsorbent.

## Results

**Synthesis and characterisation.** Inspired by the superb properties of MOFs derived from straightforward and readily-available linkers[26,27], our interest was captured by the MUF-16 series of materials. These frameworks are prepared by combining 5-aminoisophthalic acid ($H_2$aip), an inexpensive, commercially-available linker, with cobalt(II), nickel(II), or manganese(II) salts in methanol (Fig. 1a). This delivers compounds with the general formula $[M(Haip)_2]$[28,29], referred to as MUF-16 (M = Co), MUF-16(Ni) and MUF-16(Mn), respectively. These easily-handled crystalline materials are high yielding on gram scales and tolerant to oxygen and water vapour. Their crystal structures were determined by single crystal X-ray diffraction (Supplementary Table 1). The three frameworks are isostructural, belonging to the $I2/a$ space group. Individually, the metal ions adopt an octahedral geometry with four carboxylate and two amino donors arranged *trans* to one another. These ions are aligned into one-dimensional chains along a crystallographic axis supported on each side by $\mu_2$-bridging carboxylate groups (Fig. 1b). Adjacent chains are connected into two-dimensional sheets by Haip ligands that extend across the plane by coordinating to adjacent one-dimensional chains with both their amino and carboxylate donors (Fig. 1b). Only one of the two carboxyl groups of each Haip ligand coordinates to the metal. The other remains protonated and engages in hydrogen-bonding with a partner from an adjacent layer (Fig. 1c). These interactions link the layers into three-dimensional frameworks. The frameworks support one-dimensional channels of approximately $3.6 \times 7.6$ Å (accounting for the van der Waals surfaces of the atoms, Fig. 1d). In their as-synthesised form the pores contain occluded water, which can be easily removed by heating at 130 °C in vacuo.

Thermogravimetric analysis demonstrated the thermal stability of the MUF-16 materials beyond 330 °C (Supplementary Fig. S2). Their purity was established by both elemental analysis and powder X-ray diffraction (Supplementary Fig. S5). The frameworks are chemically robust, being unaffected by soaking in water or exposure to humid air for prolonged periods, as confirmed by powder X-ray diffraction and gas adsorption analysis (vide infra and Supplementary Figs. S6–S8, S13a).

As suggested by pore evident in their SCXRD structures, the MUF-16 frameworks are accessible to a range of incoming gases. Nitrogen adsorption isotherms measured at 77 K gave BET surface areas of 214, 205 and 204 m$^2$/g for MUF-16, MUF-16 (Mn), and MUF-16(Ni), respectively (Supplementary Figs. S19–S21). Total pore volumes of 0.11 cm$^3$/g were established for all three frameworks (Supplementary Table 3). These values are comparable with the geometric surface areas and pore volumes calculated from the crystallographic coordinates. The pore size distribution of MUF-16 also was calculated, which is consistent with the pore dimensions observed by SCXRD (Supplementary Fig. S12).

**Gas adsorption measurements.** $CO_2$ isotherms were collected at 293 K and up to 1 bar (Fig. 2a and see Supplementary Fig. S11 for

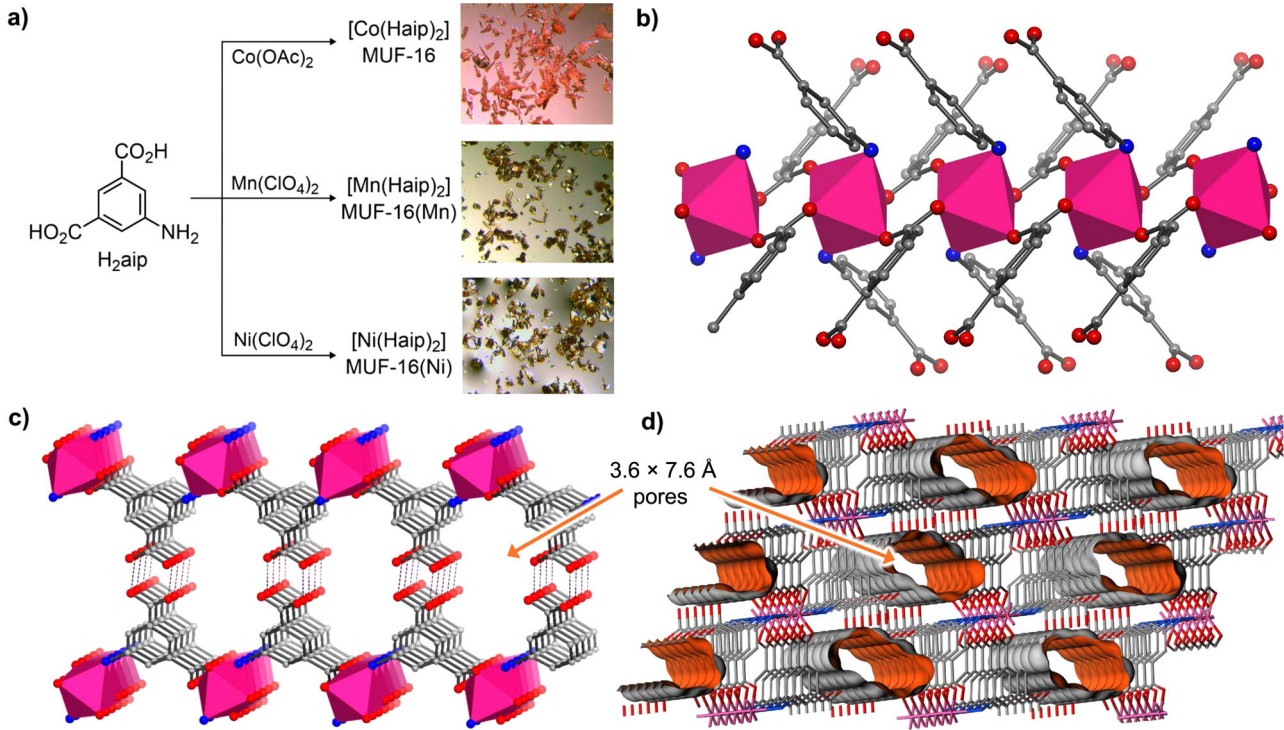

**Fig. 1 Synthesis and structure of MUF-16 materials. a** Synthetic routes to the MUF-16 family and optical micrographs of the reaction products. **b** Infinite secondary building units (iSBUs) in MUF-16 comprise one-dimensional cobalt(II) chains connected by $\mu_2$-bridging carboxylate groups of the Haip ligands (H$_2$aip = 5-aminoisophthalic acid). The cobalt(II) ions are depicted as filled octahedra. **c** The iSBUs are linked into planar two-dimensional sheets by the Haip ligands and further connected into a three-dimensional framework by hydrogen bonding (depicted as dashed lines) between adjacent sheets. **d** MUF-16 features one-dimensional channels with approximate dimensions of 3.6 × 7.6 Å that propagate through the framework. The Connolly surface of the framework is shown in orange and defined with a probe of diameter 1.0 Å. Colour code: Co = magenta; O = red; C = grey, N = blue.

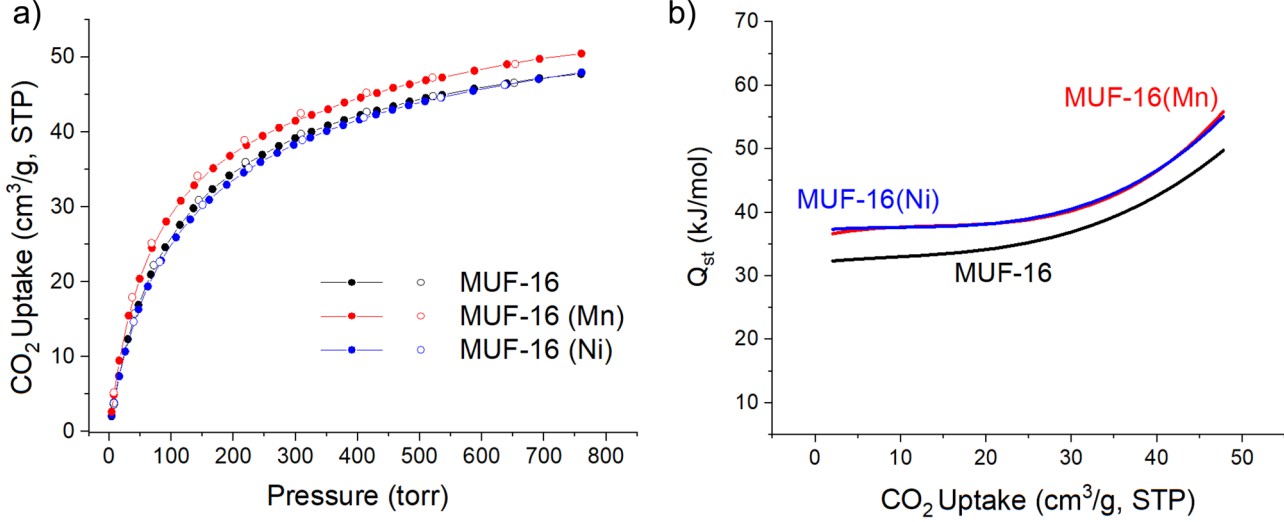

**Fig. 2 CO$_2$ adsorption on MUF-16 materials. a** Volumetric adsorption (filled circles) and desorption (open circles) isotherms of CO$_2$ at 293 K and for MUF-16 (black), MUF-16(Mn) (red), and MUF-16(Ni) (blue). **b** Heats of adsorption ($Q_{st}$) calculated for CO$_2$ binding to MUF-16 (black), MUF-16(Mn) (red), and MUF-16(Ni) (blue) as a function of CO$_2$ uptake. A high affinity for CO$_2$ coupled to a moderate heat of adsorption promise an adsorbent that takes up significant quantities of gas yet is easily recycled. Source data are provided as a Source Data file.

other temperatures). Both MUF-16 and MUF-16(Ni) take up 2.13 mmol/g (48 cm$^3$/g) at 1 bar, and MUF-16(Mn) adsorbs 2.25 mmol/g (50.5 cm$^3$/g). This equates to approximately 0.9 molecules of CO$_2$ per metal site (Supplementary Table 5). CO$_2$ uptake is only marginally higher at 273 K (Supplementary Fig. S11). The isosteric heat of adsorption ($Q_{st}$) at zero-coverage was calculated to be 32 kJ/mol for MUF-16 and 37 kJ/mol for its

Ni and Mn analogues (Fig. 2b). The $Q_{st}$ increases at higher loadings, which can be attributed to attractive intermolecular interactions when the CO$_2$ loading levels are high, which enhance the framework-CO$_2$ affinity. These interactions were experimentally verified by SCXRD (vide infra). The moderate $Q_{st}$ values, even at high CO$_2$ loading[30], are well below values observed for MOFs with open metal sites[31]. It follows that the energy

required to regenerate the frameworks by $CO_2$ desorption is likely to be low.

Single-crystal X-ray diffraction was used to identify the $CO_2$ binding sites in these frameworks[32,33]. MUF-16(Mn) was selected for this study since its darker colour streamlined crystal handling (the pale colour of the Co(II) and Ni(II) analogues make them difficult to see when loaded in a glass capillary). The results obtained for MUF-16(Mn) are directly applicable to MUF-16 and MUF-16(Ni) due to their identical structures and $CO_2$ adsorption profiles (Fig. 2a and Supplementary Fig. S5). After transferring a MUF-16(Mn) single crystal into a capillary, it was activated in vacuo and the capillary flame-sealed. This allowed the guest-free structure of MUF-16(Mn) to be determined crystallographically (Supplementary Table 2). We then filled $CO_2$ into the capillary to a pressure of 1.1 bar to determine the structure of the $CO_2$-loaded framework. We noted only minor changes to the framework itself upon evacuation and filling with $CO_2$. A clear picture of the affinity of MUF-16 for $CO_2$ arises from the $CO_2$-loaded SCXRD structure. First, the dimensions of the framework pores are well matched to the size of the $CO_2$ molecules. This allows the guests to be enveloped by multiple non-covalent contacts (Fig. 3a). Second, these contacts are favourable since the electric quadrupole of the $CO_2$ is complementary to the polarisation of the MUF-16 pore surface. For example, one of the electronegative oxygen atoms of each $CO_2$ molecule engages in N-H···O and C-H···O hydrogen bonds with framework amino and phenyl groups at distances of 2.55, 2.81, and 2.87 Å. The electropositive carbon atom of each $CO_2$ molecule engages in close-range contacts with the oxygen atoms of two non-coordinated carboxyl groups (2.87 and 3.04 Å). Two sites, which are related by crystallographic symmetry and share a common location for one of the oxygen atoms, are available to the $CO_2$ guests. They are occupied with a 50/50 ratio and refinement of the $CO_2$ occupancies gave 0.77 $CO_2$ molecules per Mn centre, which agrees with the adsorption isotherm (Supplementary Table 5) allowing for uncertainties in the exact $CO_2$ pressure in the X-ray capillaries. The $CO_2$ guest molecules are aligned along the channels and tilted with respect to the pore axis (Fig. 3b). Attractive C···O intermolecular interactions between adjacent molecules are evident at a distance of 3.78 Å. This array of $CO_2$ guests probably underlies the observed increase in $Q_{st}$ as a function of gas loading observed in the adsorption isotherms. A computational DFT model agrees with the SCXRD structure (Supplementary Fig. S60).

The strong adsorption of nitrous oxide, $N_2O$, by MUF-16 corroborates this model of $CO_2$ binding. The size and electrostatic distribution of $N_2O$ closely match those of $CO_2$ (Supplementary Fig. S9). In parallel with $CO_2$, $N_2O$ possesses atoms with partial negative charges at its termini that can bind to positively-charged regions of the pore surface, and vice-versa for its central nitrogen atom. MUF-16 adsorbs 1.91 mmol/g (43 $cm^3$/g) of $N_2O$ at 1 bar and 293 K, which is only slightly less than the uptake of $CO_2$.

The high uptake of $CO_2$ by MUF-16 contrasts with its low affinity for hydrocarbons. Adsorption isotherms of $CH_4$, $C_2H_2$, $C_2H_4$, $C_2H_6$, $C_3H_6$ and $C_3H_8$ were measured on MUF-16 at 293 K (Fig. 4a and Table 1). MUF-16 takes up just 1.20 $cm^3$/g of $CH_4$ at 1 bar and 293 K and 3.99 $cm^3$/g of $C_2H_2$. The highest adsorption amount was 5.35 $cm^3$/g observed for $C_3H_6$. Since only modest quantities of these gases are adsorbed, care was taken to ensure the accuracy of these measurements by using large sample quantities. The $Q_{st}$ values for the hydrocarbon gases are much lower than for $CO_2$ (Supplementary Table 6). The water vapour adsorption isotherm of MUF-16 was measured at 298 K, showing the steady uptake of water until saturation is reached at around two molecules per Co centre (Supplementary Fig. S13b). The isotherm is fully reversible indicating that the adsorbed water is easily removed without perturbation of the framework.

Uptake ratios provide a useful indication of the preference of an adsorbent for certain gases over others. For MUF-16, the $CO_2$/$CH_4$ uptake ratio is 39.8 (293 K and 1 bar). This is comparable to $[Cd_2L(H_2O)]$ (42.9)[34] and exceeded by only one other reported material (SIFSIX-14-Cu-i, 85) (Supplementary Table 10)[35]. Typical physisorbents show a preference for unsaturated hydrocarbons over $CO_2$, especially when bonding between the guest's π electrons and open metal sites can occur[25,36–50]. However, MUF-16 exhibits a uniform preference for $CO_2$ over all C2 and C3 hydrocarbons at 293 K and 1 bar (Table 1). Here, the uptake ratios fall between 12 (acetylene), 15.6 (ethane) and 8.9 (propene). While the limited uptake of $CH_4$ is a well-established function of its small size and low polarizability, the low affinity of MUF-16 for larger and more polar/polarizable hydrocarbon guests is notable. Inverted selectivity of this kind, that is, a preference for $CO_2$ over small hydrocarbons, is a sought after yet seldom

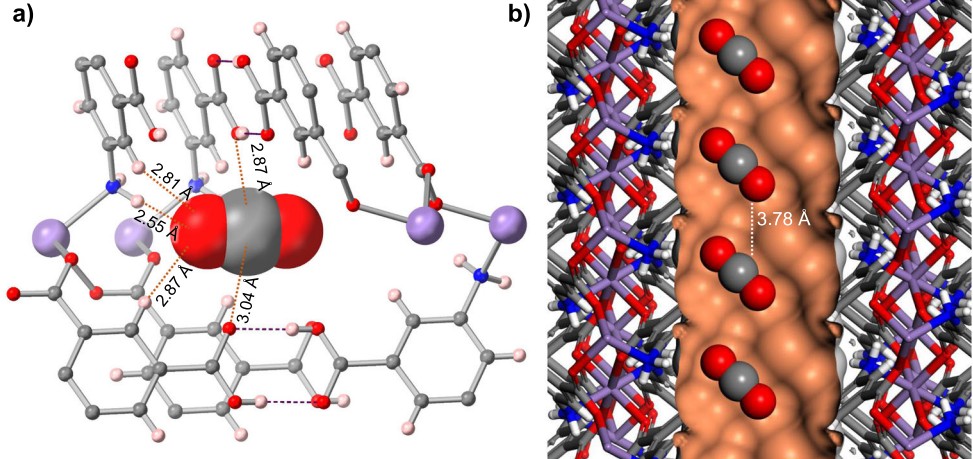

**Fig. 3 $CO_2$ capture by MUF-16. a** The adsorption sites of $CO_2$ molecules in the pores of MUF-16(Mn), as determined by single-crystal X-ray diffraction. The $CO_2$ is depicted in space-filling mode. Key intermolecular distances between MUF-16(Mn) and the adsorbed $CO_2$ are shown with dashed orange lines. A second, symmetry-equivalent $CO_2$ adsorption site exists. **b** Adsorbed $CO_2$ molecules in MUF-16(Mn) highlighting the arrangement of adsorbed $CO_2$ in the framework channels and potential attractive noncovalent interactions between adjacent guests. The $CO_2$ molecules are shown in representative orientations in one of two symmetry-related crystallographic orientations. Colour code: manganese = lilac; nitrogen = blue; oxygen = red; carbon = grey; hydrogen = pale pink or white; pore Connolly surface = orange.

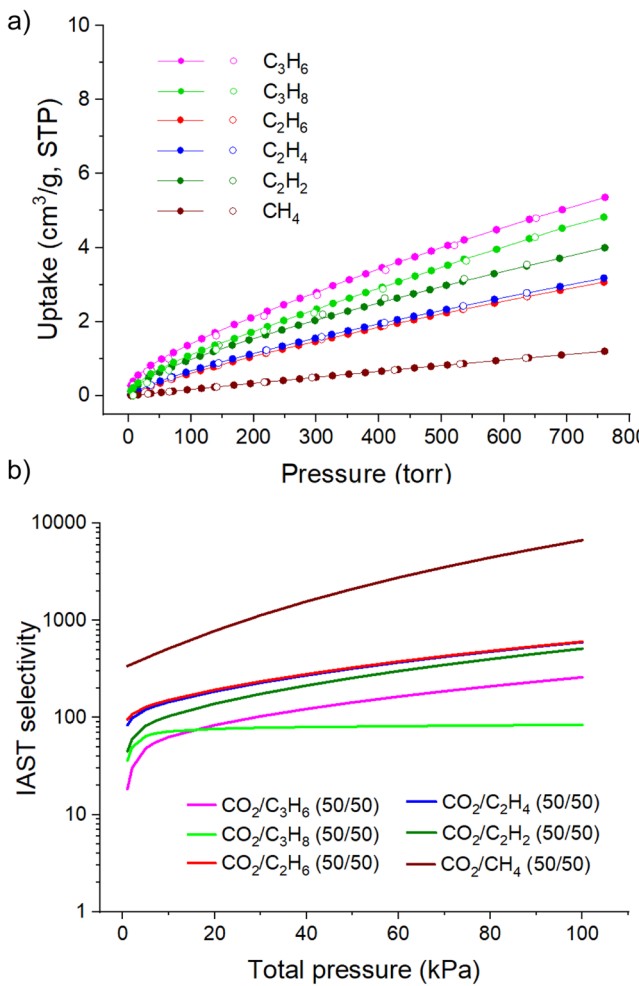

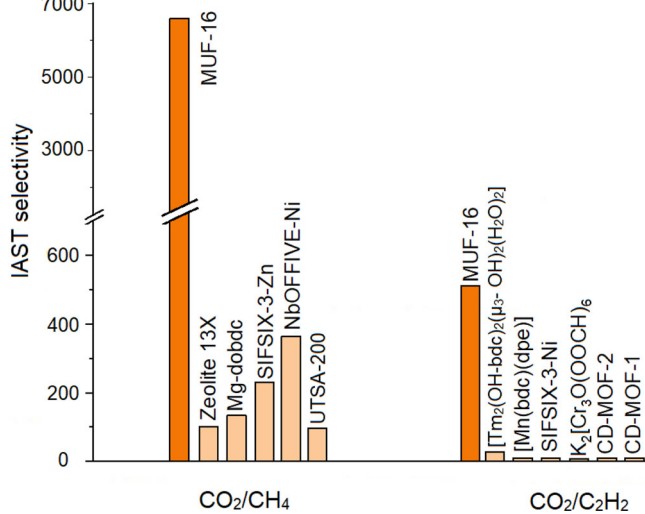

**Table 1 Summary of gas adsorption data and IAST-calculated selectivities for the MUF-16 family at 1 bar and 293 K.**

|  | MUF-16 | MUF-16(Mn) | MUF-16(Ni) |
|---|---|---|---|
| $Q_{st}$ |  |  |  |
| $CO_2$[a] | 32.3 | 36.6 | 37.3 |
| Uptake[b] |  |  |  |
| $CO_2$ | 47.78 | 50.5 | 47.97 |
| $CH_4$ | 1.20 | 3.10 | 2.77 |
| $C_2H_2$ | 3.99 | 9.69 | 7.53 |
| $C_2H_4$ | 3.17 | 8.31 | 5.42 |
| $C_2H_6$ | 3.06 | 8.81 | 5.67 |
| $C_3H_6$ | 5.35 | — | — |
| $C_3H_8$ | 4.82 | — | — |
| IAST selectivity |  |  |  |
| $CO_2/CH_4$[c] | 6690 | 470 | 1220 |
| $CO_2/C_2H_2$[c] | 510 | 31 | 46 |
| $CO_2/C_2H_4$[c] | 600 | 150 | 130 |
| $CO_2/C_2H_6$[c] | 600 | 55 | 110 |
| $CO_2/C_3H_6$[c] | 260 | — | — |
| $CO_2/C_3H_8$[c] | 84 | — | — |

[a]In kJ/mol at zero loading.
[b]In $cm^3$/g.
[c]50/50 ratio at 1 bar and 293 K as calculated by IAST.

**Fig. 4 Gas uptake and calculated separation by MUF-16. a** Experimental $CH_4$, $C_2H_2$, $C_2H_4$, $C_2H_6$, $C_3H_6$ and $C_3H_8$ adsorption (solid spheres) and desorption (open spheres) isotherms of MUF-16 measured at 293 K. **b** Predicted IAST selectivities, displayed with a log scale, of MUF-16 for various gas mixtures at 293 K. Source data are provided as a Source Data file.

**Fig. 5 Separation performance of MUF-16 compared to top-performing materials.** IAST selectivity of MUF-16 in comparison to a selection of physisorbents for $CO_2/CH_4$ (50/50) and $CO_2/C_2H_2$ (50/50) mixtures at ambient temperature and 1 bar (see Supplementary Table 11 for details). For clarity, the y axis is broken in two parts with different scales.

reported phenomenon[25,51–57]. With an uptake ratio of 12, MUF-16 surpasses previously reported materials that preferentially adsorb $CO_2$ over $C_2H_2$, including SIFSIX-3-Ni (1.2 at 298 K and 0.1 bar)[25], CD-MOF-2 (1.3 at 298 K and 1 bar)[51], $K_2[Cr_3O(OOCH)_6(4$-ethylpyridine$)_3]_2[α$-SiW$_{12}$O$_{40}]$ (4.5 at 278 K and 1 bar)[55], [Mn(bdc)(dpe)] (6.4 at 273 K and 1 bar)[52] and [Tm$_2$(OH-bdc)$_2$(μ$_3$- OH)$_2$(H$_2$O)$_2$][58] (2.8 at 298 K and 1 bar) (Supplementary Table 11). The diminished affinity of MUF-16 for $C_2H_2$ results from the reversed quadrupole moment of this guest vis-à-vis $CO_2$ (Supplementary Fig. S10). Since $C_2H_2$ is polarised oppositely to $CO_2$ it is electrostatically repelled by the functional groups that line binding pockets in MUF-16. The upshot is inverse selectivity for $CO_2$ over acetylene.

**Separations using MUF-16.** Building on the preferential affinity indicated by the uptake ratios, we quantified the selectivity of MUF-16 by Ideal Adsorbed Solution Theory (IAST) calculations[59]. At 293 K and 1 bar, the IAST selectivity of MUF-16 for $CO_2$ over $CH_4$ (50/50 mixture) is 6690 (Fig. 4b). MUF-16 is thus the best physisorbent known for this separation that does not operate by molecular sieving (Fig. 5 and Supplementary Table 10). For equimolar mixtures of $CO_2$ and $C_2H_2$, $C_2H_4$, $C_2H_6$, $C_3H_6$ or $C_3H_8$ the selectivity of MUF-16 is also high (Table 1).

With a selectivity of 510, MUF-16 is elevated well beyond other materials for the capture of $CO_2$ from $CO_2/C_2H_2$ (50/50) mixtures (Fig. 5 and Supplementary Table 11). As recognised in the literature for related systems[17,18,60], these high selectivities emerge by suppressing the uptake of the hydrocarbon gases while maintaining proficient $CO_2$ capture.

While the pore characteristics of MUF-16 clearly favour the uptake of $CO_2$ over other gases, its affinity could potentially rely on molecular sieving if the larger adsorbates are excluded from the framework on the basis of their size. This was ruled out by measuring hydrocarbon adsorption isotherms at 195 K, which showed that MUF-16 can adsorb $CH_4$, $C_2H_2$ and $C_2H_6$ (Supplementary Fig. S15). Guest molecules of this size can freely

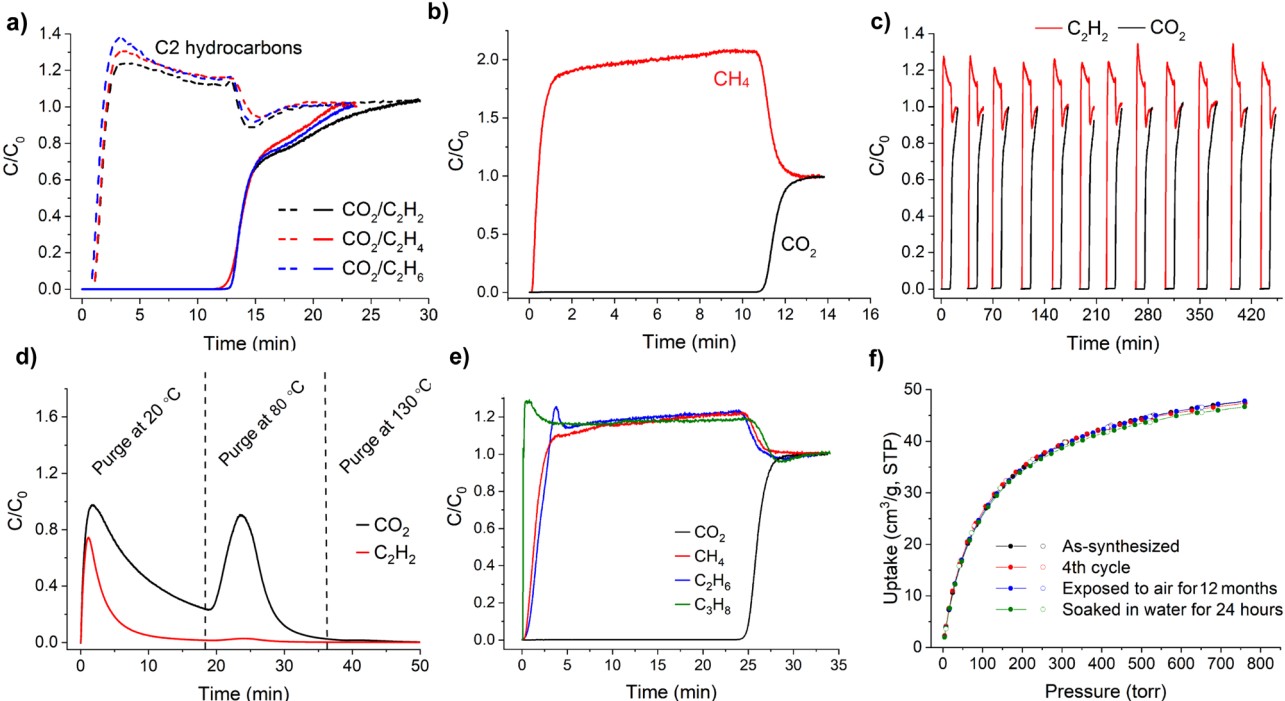

**Fig. 6 Gas separation by MUF-16. a** Experimental breakthrough curves for 50/50 mixtures of $CO_2$ and the three C2 hydrocarbons (measured independently) at 293 K and 1.1 bar in an adsorption column packed with MUF-16. **b** Experimental breakthrough curves for 50/50 mixtures of $CO_2$ and $CH_4$ at 293 K and 1.1 bar in an adsorption column packed with MUF-16. **c** Twelve separation cycles for a $CO_2$/$C_2H_2$ mixture (50/50 mixture). Each separation process was carried out at 293 K and 1.1 bar. MUF-16 was regenerated between cycles by placing it under vacuum at ambient temperature for 20–25 min. **d** Experimental desorption profile of MUF-16 following the separation of $CO_2$ and $C_2H_2$ upon heating under a helium flow of 5 $ml_N$/min at 1.1 bar. No adsorbates were removed upon further heating at 130 °C indicating that they had been fully expelled at lower temperatures. **e** Experimental breakthrough curves for a 15/80/4/1 $CO_2$/$CH_4$/$C_2H_6$/$C_3H_8$ mixture at 1.1 bar and 293 K in an adsorption column packed with MUF-16. **f** $CO_2$ adsorption isotherms (293 K) of as-synthesized MUF-16 after four consecutive adsorption-desorption cycles, after exposing it to air with ~80% humidity for 12 months, and after immersion in water for 48 hours. Source data are provided as a Source Data file.

enter the pore network of MUF-16 at this low temperature. However, since uptake is low at ambient temperatures interactions of these gases with the framework must be weak. Further, the kinetics of adsorption of several guest molecules were measured (Supplementary Fig. S16). All gases display a similar kinetic profile and reach their equilibrium uptake in well under one minute. Therefore, thermodynamic—rather than kinetic—effects have the most decisive impact on the differential affinity of these gases for MUF-16. We also considered whether a structural change of the framework might underly the gas selectivity, as observed for related systems[52]. However, XRD measurements show that the framework structure is largely conserved around room temperature in vacuo, in air and under $CO_2$ (Supplementary Fig. S6). The $CO_2$ adsorption isotherms at elevated temperatures show no sign of flexibility or gate opening (Supplementary Fig. S11), nor does the $CH_4$ isotherm at high pressure (Figure S15). In the specific case of $C_2H_4$ at 195 K, there is evidence of modest gate opening, which will be fully evaluated in future work (Figure S15).

Invigorated by these results, we then investigated the feasibility of $CO_2$/hydrocarbon separations under dynamic conditions. Experimental breakthrough curves were measured for various gas mixtures at 293 K and 1.1 bar: $CO_2$/$C_2H_6$ (50/50), $CO_2$/$C_2H_4$ (50/50), $CO_2$/$C_2H_2$ (50/50 and 5/95) and $CO_2$/$CH_4$ (50/50 and 15/85) (Fig. 6a, b; Supplementary Figs. S44 and S51). Figure 6a, b shows the dimensionless concentration of $CO_2$ and the hydrocarbons (measured independently) exiting an adsorbent bed packed with MUF-16 (0.9 gram) as a function of time.

Complete separation was realised by MUF-16, whereby the hydrocarbons broke through from the column at an early stage because of their low affinity for the framework. Conversely, the signal of $CO_2$ was not detected for at least 10 minutes due to its adsorption by MUF-16. The dynamic adsorption capacity for $CO_2$ fell in the range 1.2–1.5 mmol/g which is nearly identical to the equilibrium capacity at the relevant partial pressures of $CO_2$ (Supplementary Table 7). Significant volumes of pure hydrocarbons can be obtained in this way. Productivity calculations showed 1 kg of MUF-16 produces 27 L of the hydrocarbons from an equimolar mixture with $CO_2$ at 293 K and 1 bar. The ability of MUF-16 to selectively adsorb $CO_2$ is an important advantage of this MOF as pure hydrocarbons can be produced directly in a single adsorption stage. In literature reports to date, the capture of $CO_2$ over C2 hydrocarbons has so far largely been restricted to cryogenic temperatures and/or static conditions[52–55,57,61]. With respect to $CO_2$/$C_2H_2$ mixtures at ambient temperatures, we are aware of only a few reported materials, CD-MOF-1[51], CD-MOF-2[51] SIFSIX-3-Ni[25], and $[Tm_2(OH-bdc)_2(\mu_3- OH)_2(H_2O)_2]$[58] for which this inverse trapping of $CO_2$ has been verified by experimental breakthrough measurements. Since these MOFs adsorb $C_2H_2$ (in addition to $CO_2$) strongly at moderate pressures, their uptake ratios are modest. They are limited to very low partial pressures of $CO_2$ and suffer from low productivity.

Subsequent tests revealed that MUF-16 maintains its $CO_2$ uptake and the complete removal of $CO_2$ over at least 12 separation cycles (Fig. 6c). MUF-16 was regenerated between cycles by placing it under vacuum or by purging with an inert gas

(Fig. 6d). Virtually all of the adsorbed acetylene and around half of the $CO_2$ can be removed from the bed by purging at room temperature. The remainder can be fully desorbed at 80 °C.

To investigate separations involving trace $CO_2$, we simulated breakthrough curves of feed gases with low $CO_2$ partial pressures. First, a mass transfer coefficient was empirically determined based on measured breakthrough results to produce a match between simulated and experimental breakthrough curves[26,62]. With this realistic mass transfer coefficient in hand, we predicted breakthrough curves using feeds containing 0.1% $CO_2$ in $C_2H_2$ (Supplementary Fig. S57). These calculations revealed that MUF-16 can eliminate trace quantities of $CO_2$, as often required in industrial processes.

We then turned our attention to the separation of more complex gas mixtures. MUF-16 captures the $CO_2$ from $CO_2/CH_4/C_2H_6/C_3H_8$ (15/80/4/1) feed mixtures at 1.1 bar. Here, we observed $CH_4$, $C_2H_6$ and $C_3H_8$ to break through quickly with steep elution profiles (Fig. 6e). Crucially, the larger $C_2H_6$ and $C_3H_8$ components do not diminish the $CO_2$ capture capabilities of MUF-16. This is an important observation for the removal of $CO_2$ from natural gas, where mixed-gas separations involving these hydrocarbons are often required yet the pool of competent materials is limited[19,63]. To further probe the applicability of MUF-16 to natural gas sweeting, we conducted breakthrough measurements at a higher pressure of 9 bar. $CO_2$ was cleanly removed from the gas stream (Supplementary Figs. S45 and S46). Breakthrough simulations at pressures relevant to natural gas processing (50 bar) lead to the prediction that MUF-16 can capture $CO_2$ from natural gas (Supplementary Fig. S50). Water vapour is a component of crude natural gas streams and it can affect gas adsorption by physisorbents[64,65]. To test the moisture resistance of MUF-16, we measured its $CO_2$ adsorption properties after exposure to air and immersion in water (Fig. 6f). The framework retains its $CO_2$ adsorption capacity following these mistreatments. More detailed analysis, including the impact of water vapour on gas separation and the resistance of MUF-16 to other common natural gas impurities such as $H_2S$, is an important next step.

In summary, the pores in MUF-16 are complementary to $CO_2$ in size and electrostatic potential. This allows H-bonding and other noncovalent interactions to trap the guest $CO_2$. Other guests, specifically methane and the $C_2$ hydrocarbons, do not bind efficiently. This arises from the reversed polarity of these guests with respect to $CO_2$ and results in a strong preference for $CO_2$ over methane and inverted selectivity for $CO_2$ over $C_2$ and $C_3$ hydrocarbon guests. MUF-16 shows exceptional performance for $CO_2/CH_4$ and $CO_2/C_2H_2$ separations across a range of $CO_2/$hydrocarbon compositions and pressures. These observations are relevant to the practical challenges of purifying natural gas and industrial feedstocks. MUF-16 has the potential to be produced economically on large scales and its chemical stability and recyclability meet the demands of a long-lived physisorbent. Given these characteristics, MUF-16 is a promising physisorbent for the capture of $CO_2$.

## Data availability

Source data are provided with this paper. Crystallographic data and files of MUF-16 as synthesized, under vacuum and loaded with $CO_2$ have been deposited (CCDC 1948901 - 1948905). Additional graphics, TG curves, PXRD diffractograms, multiple cycle adsorption isotherms, dual site Langmuir isotherm model fitting, isosteric heat of adsorption calculations, BET surface area calculations, IAST calculations of adsorption selectivities, breakthrough curves simulations and models used and column breakthrough test setup with procedures and measurements, and the DFT results are available as Supplementary Information.

Further data that support the findings of this study are available from the corresponding author upon reasonable request. Source data are provided with this paper.

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

## Acknowledgements

We would like to thank Victoria-Jayne Reid for technical assistance, Seok June (Subo) Lee and Adil Alkas for useful discussions regarding X-ray crystallography, and Steve Denby for expert engineering support. We gratefully acknowledge the MacDiarmid Institute and RSNZ Marsden Fund (contract 14-MAU-024) for financial support. R.B. acknowledges the National Computing Infrastructure (NCI) for providing the super-computing facility.

## Author contributions

The manuscript was written through the contributions of O.T.Q. and S.G.T. who designed and performed the experiments, analysed the results and jointly wrote the paper. R.B. performed the DFT calculations. A patent on MUF-16 has been lodged (WO 2020/130856 A1).

## Competing interests

The authors declare no competing interests.
