## [Peer Review File · Nature Communications]

REVIEWER COMMENTS

Reviewer #1 (Remarks to the Author):

This work reports a rare CO₂-selective adsorption of a two-dimensional MOF with modest pore size ($3.6 \times 7.6 \text{ \AA}^2$), that shows only CO₂ adsorption under ambient conditions. This MOF has been fully explored for separation in this manuscript, especially shown in various breakthrough experiments. It is noted that MUF-16 can simultaneously adsorb CO₂, C₂H₂ and C₂H₆ at 195 K in contrast to minor adsorption of C₂H₂ at ambient temperature, which might indicate a typical gate-opening (threshold adsorption) in flexible MOFs. Such adsorption phenomenon has been reported in a Mn MOF (J. Am. Chem. Soc. 2016, 138, 3022) that also shows large CO₂ adsorption but minor C₂H₂ sorption under ambient conditions, while adsorb both gases at cryogenic temperature. The potential flexibility is quite reasonable to illustrate such sorption behavior especially for a 2D MOF. The current speculation cannot well illustrate such adsorption difference, that is a significant drawback of this work. Especially the speculated electrostatic repulsions cannot be applied for explaining the C₂H₂ adsorption at 195 K. This work is of great interesting for relevant research communities, but this major issue need to be addressed before the acceptance.

In Figure S15, only CO₂, C₂H₂ and C₂H₆ sorption isotherms at 195 K were provided. How about that for C₂H₄?

Reviewer #2 (Remarks to the Author):

This work reported by Telfer et al. described the synthesis of a family of MOFs (MUF-16) and their gas adsorption and separation properties. The suitable pore size and pore environment of this family of MOF have enabled record-setting CO₂/CH₄ and CO₂/C₂H₂ selectivities. The CO₂/C₂H₂ separation is especially interesting since most MOFs prefer to adsorb C₂H₂ and very few MOFs are known to have this inverse selectivity. Moreover, the selectivity here is substantially higher. The authors can also determine the adsorption sites using single crystal X-ray diffraction which helps with the understanding of adsorption mechanism. This is a very interesting work that represents the state-of-the-art study in CO₂/CH₄ and CO₂/C₂ separations. The following comments should be addressed during the revision.

1. Page 1, "In the search of effective MOF physisorbents, it is clear that simply searching for materials with ever-higher levels of CO₂ uptake per se is unlikely to produce an adsorbent that is adept at gas separations. Instead, significant advances will emerge by suppressing the uptake of competing gases,"

The above statement seems to over-emphasize selectivity or the ability of researchers to control selectivity as compared to capacity. I hope this statement can be revised to balance the emphasis to both selectivity and capacity, by citing related works that highlight the importance of uptake capacity as well (ref. 24 should be cited here as well, plus other latest studies such as J. Am. Chem. Soc. 2020, 142, 2222).

2. "The moderate Q_{st} values, even at high CO₂ loading, are well below values observed for MOFs with open metal sites²⁸⁹⁸. It follows that the energy required to regenerate the frameworks by

CO₂ desorption is likely to be low."

Given the applications targeted here are very general and can be potentially achieved with any MOF type, the comparison should be made to all MOFs, not just MOFs with OMS. Especially, the pacs-type MOFs have low Q_{st} and should be compared to for completeness (Acc. Chem. Res. 2017, 50, 407).

3. Table S1, why 2H₂O for Co, and 3H₂O for Mn and Ni?

4. Also Table S1, why the a and c axes are switched for Mn? If they are isostructural, should use the consistent unit cells.

5. The unit cell of Mn in Table S1 is not consistent with those in Table S2. The a and c axes are switched.

6. Table S10, the header contains CO₂/N₂/CH₄, but the table only has CO₂/CH₄.

Reviewer #3 (Remarks to the Author):

In this manuscript, Telfer and coworker report the separation of CO₂ from light hydrocarbons (C₁-C₃) by a series of MOFs, namely M(Haip)₂ (M = Co, Mn, Ni, H₂aip = 5-aminoisophthalic acid). These compounds are not new - they were previously reported by other groups (Refs. #26-27), but the separation performance, particularly the preferred adsorption of CO₂ over C₂H₂ with high selectivity, is attractive, and may be potentially useful in practical applications. It may become suitable for publication in Nature Communications if the following questions can be well addressed.

1) The 3D framework is formed through hydrogen bonding, rather than coordinate connection, between adjacent sheets. The authors claim the material is stable beyond 330 °C based on TG plot (Fig. S2). This is not convincing/sufficient, as the onset temperature of weight loss is not necessarily an indicator for crystallinity. PXRD measurements must be carried out on the sample after heating at this temperature to confirm its framework stability.

2) The heats of adsorption increase at higher loadings for CO₂ (Figure 2b) which is quite unusual. This was explained by the attractive C···O intermolecular interactions between adjacent molecules at a distance of 3.78 Å, which is not clear to me as the adsorption is monolayer, and the close contacts between CO₂ and the pore surface are the dominating driving force for adsorption. I suggest to calculate the strength of interaction in a different way, say using direct Clausius equation to confirm the values calculated from virial fitting. Experimentally, calorimetric measurements would give solid results.

3) It is interesting that the materials adsorb negligible C₂H₂, in contrast to CO₂. This was attributed to the low affinity to C₂H₂, and electropositive regions around the C₂H₂ termini may induce repulsive interactions with the framework pore surface (Figure S10). The orientation of C₂H₂ in the pores (see Figure S10) should be experimentally confirmed by single crystal X-ray diffraction (higher occupancy may be possible at lower temperature). The orientation could be different from CO₂ molecules.

4) The materials are robust in humid air, as confirmed experimentally. Do activated samples adsorb water? Would water be a competing adsorbate during CO₂/C₂H₂ separation? Water adsorption

isotherms and selectivities should be provided and results discussed in the manuscript. If the materials do adsorb water, breakthrough measurements under humid conditions need to be carried out and performance compared with those at dry conditions.

Reviewer #4 (Remarks to the Author):

The communication by Teifer et al report a study on a particular MOF with a title "Selective Capture of Carbon Dioxide from Hydrocarbons Using a Metal-Organic Framework: Relevance to the Purification of Natural Gas and Acetylene".

The work in all respect except the single crystal X-ray study appears to be expertly executed. The work very heavily relies on the X-ray structures of the MOF's, particularly on the Mn-MOF with CO₂ molecule modelled/found inside. What the authors have done is to use the MOF as a crystalline sponge to capture CO₂. Very surprisingly the authors do not cite the work by Makoro Fujita on crystalline sponges and the problems encountered in their single crystal X-ray studies. The non-covalently trapped guest(s) inside a crystalline sponge (MOF) are extremely difficult to reliably detect. Now the authors have modelled a disordered CO₂ molecule at RT (296 K). The thermal displacement parameters are 10 times higher than within the MOF frame. It is very likely that the occupancy of the CO₂ is much less than now given 1 (2 x 0.5). This referee does not believe that the CO₂ can be located in this quality data.

Thus this work is premature for publication in Nat. Comm. or in any journal before the X-ray work is properly done or removed from the ms.

The authors must:

1. Properly read and cite M. Fujita work on the use of crystalline sponges (MOFs), e.g. IUCrJ 3 (2016), 139 - 151.
2. Do the XRD work at 100 K on several samples and only if the occupancy of the CO₂ is high enough with reasonable thermal parameters then use it as basis of the discussion.

Reviewer #1 (Remarks to the Author):

This work reports a rare CO₂-selective adsorption of a two-dimensional MOF with modest pore size (3.6 × 7.6 Å²), that shows only CO₂ adsorption under ambient conditions. This MOF has been fully explored for separation in this manuscript, especially shown in various breakthrough experiments. It is noted that MUF-16 can simultaneously adsorb CO₂, C₂H₂ and C₂H₆ at 195 K in contrast to minor adsorption of C₂H₂ at ambient temperature, which might indicate a typical gate-opening (threshold adsorption) in flexible MOFs. Such adsorption phenomenon has been reported in a Mn MOF (J. Am. Chem. Soc. 2016, 138, 3022) that also shows large CO₂ adsorption but minor C₂H₂ sorption under ambient conditions, while adsorb both gases at cryogenic temperature. The potential flexibility is quite reasonable to illustrate such sorption behavior especially for a 2D MOF. The current speculation cannot well illustrate such adsorption difference, that is a significant drawback of this work. Especially the speculated electrostatic repulsions cannot be applied for explaining the C₂H₂ adsorption at 195 K. This work is of great interesting for relevant research communities, but this major issue need to be addressed before the acceptance.

We thank the referee for drawing our attention to the flexibility of the Mn MOF (as reported in (J. Am. Chem. Soc. 2016, 138, 3022) and the way that this structural change might explain the CO₂/C₂H₂ selectivity exhibited by this framework. While our experimental results did not indicate any gate opening, we have used this as an opportunity to further consider (i) whether MUF-16 is flexible, and (ii) whether this flexibility underlies the observed gas selectivity.

Firstly, is MUF-16 flexible? Despite conducting substantial experimental work on this MOF, we saw no evidence for flexibility *until* we measured its C₂H₄ isotherm at 195 K (as part of these revisions to address part 2 of this referee's comment). The C₂H₄/195K isotherm is consistent with framework gate opening at around 280 Torr. This leads to a modest increase in gas uptake (~15%). No flexing is observed for C₂H₄ at 273 K or 293 K or above nor for CH₄, C₂H₆ and C₂H₂, as originally reported in the manuscript.

Second, is a structural rearrangement of MUF-16 responsible for its selectivity? In other words, do the CO₂ and non-CO₂ guests "see" different structures? If this were the case, the structure of the framework with CO₂ guests should be different to the structure without CO₂ guests (that is, *in vacuo* or loaded with different molecules). This putative structural change should be manifest by XRD. Since there is no sign of a change in structure (details below) any framework distortion must be slight. A second line of evidence comes from adsorption isotherms: Gate-opening and related changes lead to abrupt increases in gas adsorption, which are easily distinguished from typical smooth isotherms (using a log scale on the pressure axis if necessary). The only isotherm to exhibit this kind of behaviour was the newly-measured C₂H₄/195K isotherm.

Our conclusion from the experimental evidence is that framework flexing is not a general phenomenon and does not underlie the observed selectivity. We note that the C₂H₄/195K isotherm is a special case with gate opening around 280 Torr and a minor increase in uptake. Gate opening is not observed under other conditions so it cannot be responsible for the observed selectivity. This phenomenon will be investigated more deeply in follow-up work.

Our detailed reasoning follows.

- The CO₂ loaded framework structure must be different to both the *in vacuo* structure and after loading with non-CO₂ guests. However, a series of experiments point to all frameworks having the same structure. Specifically,

- The SCXRD structures of measured *in vacuo* and under 1 bar of CO₂ are virtually identical (Table S2). An overlay of these structures is shown below.

- The PXRD patterns of MUF-16 sealed in a capillary and measured (i) under vacuum, and (ii) under CO₂ (1 bar) are very similar, indicating little (if any) structural change. These new data are presented in Fig S6.
- The CO₂ isotherms show no sign of flexibility or gate opening (Fig S11) in the temperature range 270 – 390 K. The isotherms, even when viewed on a log scale (this log plot has been newly added to Fig S11), are smooth and continuous.
- There is no indication of flexing in the high pressure CH₄ isotherm (measured at 293 K up to 125 bar). The data are now presented in Fig S15.

These experiments show that any flexibility is subtle and does not involve a large structural rearrangement. If the flexing is to impact on the selectivity towards CO₂ then the framework must flex at very low temperatures around room temperature as soon as any CO₂ is introduced. However, there is no evidence for this in the isotherms measured between 270 – 390 K nor in the comparisons of the relevant XRD structures and patterns. We conclude that a structural rearrangement of MUF-16 does not underpin its selectivity. That is not to say that MUF-16 is not flexible at all – we have discovered that it is flexible in certain situations and we will follow up on this finding in due course.

To add clarity on this point to the manuscript, we now write “We also considered whether a structural change of the framework might underlie the gas selectivity, as observed for related systems. However, the XRD measurements show that the framework structure is largely conserved around room temperature *in vacuo*, in air and under CO₂ (Fig S6). The CO₂ adsorption isotherms at elevated temperatures show no sign of flexibility or gate opening (Fig. S11), nor does the CH₄ isotherm at high pressure (Fig. S15). In the specific case of C₂H₄ at 195 K, there is evidence of modest gate opening, which will be fully evaluated in future work.”

In Figure S15, only CO₂, C₂H₂ and C₂H₆ sorption isotherms at 195 K were provided. How about that for C₂H₄?

This oversight was rectified by measuring the C₂H₄ isotherm at 195 K and including the data in Fig. S15.

Reviewer #2 (Remarks to the Author):

1. Page 1, "In the search of effective MOF physisorbents, it is clear that simply searching for materials with ever-higher levels of CO₂ uptake per se is unlikely to produce an adsorbent that is adept at gas separations. Instead, significant advances will emerge by suppressing the uptake of competing gases,"

The above statement seems to over-emphasize selectivity or the ability of researchers to control selectivity as compared to capacity. I hope this statement can be revised to balance the emphasis to both selectivity and capacity, by citing related works that highlight the importance of uptake capacity as well (ref. 24 should be cited here as well, plus other latest studies such as J. Am. Chem. Soc. 2020, 142, 2222).

In this section we are highlighting the importance of selectivity rather than outright capacity. The search for frameworks with ever higher capacities is likely to be fruitless for producing a useful adsorbent if the adsorption of the non-CO₂ component also rises. As written, we certainly allow for high capacity MOFs as long as this capacity is couple to high selectivity. So, we are certainly on the same page as the referee. We have modified this text in this part to make the situation clearer:

"In the search of effective MOF physisorbents, simply searching for materials with ever-higher levels of CO₂ uptake per se may not deliver adsorbents that are adept at gas separations since the adsorption of non-CO₂ components may also rise. Instead, significant advances will emerge by suppressing the uptake of these competing gases,⁹ developing scalable synthetic protocols, mitigating the impact of common impurities such as water vapour and oxygen, and developing low energy pathways to adsorbent recycling."

We note that this statement comes in the introduction and serves to paraphrase existing literature (ref 9) rather than introducing our own ideas.

2. "The moderate Q_{st} values, even at high CO₂ loading, are well below values observed for MOFs with open metal sites²⁸⁹⁸. It follows that the energy required to regenerate the frameworks by CO₂ desorption is likely to be low."

Given the applications targeted here are very general and can be potentially achieved with any MOF type, the comparison should be made to all MOFs, not just MOFs with OMS. Especially, the pac-type MOFs have low Q_{st} and should be compared to for completeness (Acc. Chem. Res. 2017, 50, 407).

This citation has been added to the manuscript. Just for clarity, we place emphasis on frameworks with OMS since they tend to have high Q_{st} values but suffer from a high energy penalty in the regeneration step (whereas there are indeed thousands of MOFs without OMS).

3. Table S1, why 2H₂O for Co, and 3H₂O for Mn and Ni?

This reflects the water molecules that occupied discrete positions in the pores and thus could be located by SCXRD. The cobalt material likely has a third water molecule in the pore but disordered to the extent that it cannot be pinpointed by SCXRD.

4. Also Table S1, why the a and c axes are switched for Mn? If they are isostructural, should use the consistent unit cells.

We respectfully point out that this makes no material difference to the structures or the analysis so we are happy to remain with this designation of the axes.

5. The unit cell of Mn in Table S1 is not consistent with those in Table S2. The a and c axes are switched.

As above – there is no material difference and the choices of axes is arbitrary and they can be interchanged by anyone familiar with crystallography. The two structures compared in Table S2 have an identical choice of axis.

6. Table S10, the header contains CO₂/N₂/CH₄, but the table only has CO₂/CH₄.

We thank the referee for pointing out this oversight, which has now been corrected.

Reviewer #3 (Remarks to the Author):

1) The 3D framework is formed through hydrogen bonding, rather than coordinate connection, between adjacent sheets. The authors claim the material is stable beyond 330 °C based on TG plot (Fig. S2). This is not convincing/sufficient, as the onset temperature of weight loss is not necessarily an indicator for crystallinity. PXRD measurements must be carried out on the sample after heating at this temperature to confirm its framework stability.

We thank the referee for their astute comments and have added a photo (Fig S2) and PXRD pattern of MUF-16 (Fig S6) after heating to 330 deg C in the TGA. In both cases the MUF-16 sample resembles pristine material, showing that the framework is stable to this temperature.

2) The heats of adsorption increase at higher loadings for CO₂ (Figure 2b) which is quite unusual. This was explained by the attractive C···O intermolecular interactions between adjacent molecules at a distance of 3.78 Å, which is not clear to me as the adsorption is monolayer, and the close contacts between CO₂ and the pore surface are the dominating driving force for adsorption. I suggest to calculate the strength of interaction in a different way, say using direct Clausius equation to confirm the values calculated from virial fitting. Experimentally, calorimetric measurements would give solid results.

We thank the referees for these comments and the opportunity to present Q_{st} data by direct use of Clausius-Clapeyron relationship. The results are virtually identical to the method fitted with virial method:

Isosteric heats of adsorption were determined using the integrated form of the Clausius-Clapeyron equation by calculating the slope of ln(P) vs 1/T for each loading (amount adsorbed):

$$\left[\frac{\partial \ln P}{\partial \left(\frac{1}{T} \right)} \right]_q = - \frac{Q_{st}}{R}$$

Where Q_{st} is isosteric heat of adsorption, q is gas uptake and R is gas constant. A line is fit to a plot of $1/T$ vs $\ln P$ for each loading with the slope affording Q_{st}/R . An error in the isosteric heat for a given loading can be calculated from the standard error in the slope of the best-line. It is important to note that the Clausius–Clapeyron equation assumes that the isosteric heat of adsorption does not vary with temperature. We chose nine different uptakes three temperatures (293, 298 and 303 K) to calculate the Q_{st} for CO_2 binding.

Model	linear (User)								
Equation	(-A*x)+B								
Plot	2 cm ³ /g	7 cm ³ /g	12 cm ³ /g	17 cm ³ /g	22 cm ³ /g	27 cm ³ /g	32 cm ³ /g	37 cm ³ /g	42 cm ³ /g
A	3445.78881 ±	3485.38214 ±	3501.34642 ±	3596.48326 ±	3714.23536 ±	3942.36188 ±	4155.8022 ±	4591.73045 ±	5466.71326 ±
B	13.17642 ± 1.	14.66728 ± 1.3	15.37621 ± 0.8	16.17616 ± 1.	16.98405 ± 1.	18.1352 ± 1.18	19.24305 ± 0.	21.13944 ± 0.8	24.60065 ± 0.8
Reduced	8.73949E-4	0.00109	3.74654E-4	6.90338E-4	6.21992E-4	7.93419E-4	5.58079E-4	4.44445E-4	3.6969E-4
R-Square	0.98853	0.98609	0.99521	0.99166	0.99294	0.99202	0.99493	0.99669	0.99805
Adj. R-Sq	0.97706	0.97217	0.99041	0.98332	0.98589	0.98404	0.98987	0.99338	0.99611

Linear fitting for the Clausius-Clapeyron equation to calculate heat of adsorption using three temperatures of 293, 298 and 303 K.

Isosteric heat of adsorption of CO_2 for MUF-16 calculated by direct use of Clausius-Clapeyron equation.

As depicted in Fig 3b of the manuscript, vdW interactions between adsorbed CO₂ molecules become more important at high guest loading so guest-guest interactions add to the host-guest interactions. In essence, the adsorption of CO₂ creates pockets that encourage the further adsorption of CO₂, which no longer relies simply on interactions with the pore wall. Relevant literature can be found here: [10.1021/ja076595g](https://doi.org/10.1021/ja076595g). We have reworded the manuscript to make this point clearer to now read “The Q_{st} increases at higher loadings, which can be attributed to attractive intermolecular interactions when the CO₂ loading levels are high, which enhance the framework-CO₂ affinity.”

3) It is interesting that the materials adsorb negligible C₂H₂, in contrast to CO₂. This was attributed to the low affinity to C₂H₂, and electropositive regions around the C₂H₂ termini may induce repulsive interactions with the framework pore surface (Figure S10). The orientation of C₂H₂ in the pores (see Figure S10) should be experimentally confirmed by single crystal X-ray diffraction (higher occupancy may be possible at lower temperature). The orientation could be different from CO₂ molecules.

Absolutely, the orientation of the C₂H₂ guests will most certainly be different to that of CO₂! That is exactly the point we were trying to make, so our original wording was not sufficiently precise. The C₂H₂ will have to occupy a different position/orientation compared to the CO₂, as will all of the other hydrocarbon guests. The text has now been altered to read “The diminished affinity of MUF-16 for C₂H₂ may result from the reversed quadrupole moment of this guest vis-à-vis CO₂, as illustrated by a hypothetical loading model (Figure S10). This is only overcome for C₂H₂ and the other hydrocarbon guests at reduced temperatures and by binding in a different location to that observed for CO₂.”

Unfortunately, we don't have technical capabilities to load a high pressure of C₂H₂ (or any other gas) at low T so establishing their binding locations is beyond our reach. It would certainly be unusual to try and pinpoint the binding site of a weakly-adsorbed guest, at any rate.

4) The materials are robust in humid air, as confirmed experimentally. Do activated samples adsorb water? Would water be a competing adsorbate during CO₂/C₂H₂ separation? Water adsorption isotherms and selectivities should be provided and results discussed in the manuscript. If the materials do adsorb water, breakthrough measurements under humid conditions need to be carried out and performance compared with those at dry conditions.

We thank the referee for their comments. Unlike flue gas, for example, H₂O is not relevant to the acetylene/CO₂ separation: no other papers in the literature investigate water uptake related to this application. The measurement of water uptake isotherms and competitive breakthrough curves with humidity is also a significant and time-consuming undertaking. It will be the subject of future investigations and reported comprehensively in subsequent manuscripts. We have to draw a line somewhere and we already state “More detailed analysis, including the resistance of MUF-16 to other common natural gas impurities such as H₂S, is an important next step.” We have now modified this to read “More detailed analysis, including the impact of water vapor on gas separation and the resistance of MUF-16 to other common natural gas impurities such as H₂S, is an important next step.”

The current manuscript cannot fairly be described as lightweight – it already spans from synthesis to breakthrough studies and is detailed in a thorough and comprehensive manner. Any further experimental work, however valuable, will provide additional information rather than comprising an essential component of this study. Reflecting this, the word limit for Nature Comms is 5000 and our manuscript already exceeds 6450 words.

Reviewer #4 (Remarks to the Author):

The work in all respect except the single crystal X-ray study appears to be expertly executed. The work very heavily relies on the X-ray structures of the MOF's, particularly on the Mn-MOF with CO₂ molecule modelled/found inside. What the authors have done is to use the MOF as a crystalline sponge to capture CO₂. Very surprisingly the authors do not cite the work by Makoro Fujita on crystalline sponges and the problems encountered in their single crystal X-ray studies. The non-covalently trapped guest(s) inside a crystalline sponge (MOF) are extremely difficult to reliably detect. Now the authors have modelled a disordered CO₂ molecule at RT (296 K). The thermal displacement parameters are 10 times higher than within the MOF frame. It is very likely that the occupancy of the CO₂ is much less than now given 1 (2 x 0.5). This referee does not believe that the CO₂ can be located in this quality data.

Thus this work is premature for publication in Nat. Comm. or in any journal before the X-ray work is properly done or removed from the ms.

The authors must:

1. Properly read and cite M. Fujita work on the use of crystalline sponges (MOFs), e.g. IUCrJ 3 (2016), 139 - 151.
2. Do the XRD work at 100 K on several samples and only if the occupancy of the CO₂ is high enough with reasonable thermal parameters then use it as basis of the discussion.

We thank the referee for their encouragement to read widely on the subject of guest encapsulation in framework materials. Indeed, we are already intimately familiar with the literature here. We did, however, fail to cite relevant literature on the use of XRD to locate small molecular guests in MOFs. We have now remedied that oversight (new references 30 and 31).

We respectfully note that the isotherms show that the pores are nearly fully occupied (i.e. they approach saturation) at around room temperature and one bar. Thus, the occupancy of the CO₂ in the crystal can barely be raised beyond the point at which we have measured. Nonetheless, the trapped CO₂ clearly revealed itself in the Fourier difference map of our original structure, and the positions of the CO₂ atoms were allowed to refine freely and they converged without difficulty.

There are >1800 hits for a CSD search of molecular CO₂ in the presence of a metal, most of which are MOFs. Several hundred of these structures involve data measurements around RT. Identifying CO₂ adsorption sites in MOFs by XRD at room temperature is thus a routine and commonly-accepted method. In the case of MUF-16 it was a deliberate strategy on our part to measure the SCXRD at RT to highlight that CO₂ binding readily takes place around RT i.e. low temperatures are not required to induce guest adsorption.

When XRD measurements are made around RT thermal displacement parameters of the CO₂ are far larger than those of the framework. This is expected since the CO₂ is a loosely-held guest molecule. The higher displacement parameters are reflective of its anticipated thermal mobility. Literature examples of this phenomenon appear, *inter alia*, in *Chemical Science* (2019, 10, 10018-10024), *Chem. Commun.* (2018, 54, 4262-4265), and *Nat. Commun.* (2017, 8, 14085).

Paper	SCXRD temp	Average framework displacement parameter	CO₂ displacement parameters	CO₂ treatment
Chemical Science 2019	RT	0.050	0.30 (average)	isotropic
Chem. Commun. 2018	RT	0.045	0.25 (fixed)	isotropic
Nat. Commun. 2017	RT	0.04	0.316 (fixed)	isotropic

Furthermore, the quality of the reported dataset of MUF-16(Mn)-CO₂ is acceptably high ($R_{\text{int}} = 0.1104$). Therefore, remeasuring the data at a lower temperature would not be warranted even if we reconsidered the philosophy behind this experiment. We also note that the raw XRD data for this structure are available as ESI.

In light of the referee's comments, we have revisited the structure of the CO₂-loaded MUF-16 to address their point that "It is very likely that the occupancy of the CO₂ is much less 1." Instead of holding the occupancy fixed at 1, we allowed it to freely refine. It converged on a high occupancy (0.77), so we have included these new data in place of the original. The text in the main manuscript has been updated to read: "They are occupied with a 50/50 ratio and refinement of the CO₂ occupancies gave 0.77 CO₂ molecules per Mn centre, which agrees with the adsorption isotherm (Table S5) allowing for uncertainties in the exact CO₂ pressure in the X-ray capillaries." The data in the ESI has also been updated in accord with this new structure.

REVIEWER COMMENTS

Reviewer #1 (Remarks to the Author):

This reviewer cannot be convinced by their responses in terms of framework flexibility and gate pressures. Quite a lot parameters will affect their gate pressures including the temperature, pressure, and examined gas molecules. Without clear understanding, the mechanism for this gas separation cannot be rationalized. Current explanation for this gas separation is not reasonable. This work is not ready for the publication yet in NC.

Reviewer #2 (Remarks to the Author):

The authors have addressed the questions I raised, and I am ok with the current form.

Reviewer #3 (Remarks to the Author):

The authors have addressed most of my questions except the request for water adsorption experiment. I still think it is important to carry out this experiment especially because the MOF is very hydrophilic. The experiment is fairly simple and not time consuming. I also noted Ref. #48 is missing part of the information.

Reviewer #5 (Remarks to the Author):

The manuscript entitled "Selective Capture of Carbon Dioxide from Hydrocarbons Using a Metal-Organic Framework: Relevance to the Purification of Natural Gas and Acetylene" by Omid et al. reports the selective capture of CO₂ from hydrocarbons using MUF-16 (Mn). The manuscript is reviewed with special attention to the crystallographic part as per the editor's request. The overall manuscript is well written, and experiments have been performed to high standards. I recommend this manuscript to be accepted for publication in Nat. Comm. given the authors can address the following crystallographic concerns.

(1) The crystallographic work showing the presence of CO₂ inside the MOF pores is convincing but given the lower occupancy (0.77) and large isotropic displacement factors it is not recommended to discuss any CO₂-based geometrical parameters quantitatively. Instead, the authors should employ density functional theory-based calculations to complement their experimental data. This should be a straightforward process using the experimental geometry data as starting point. The resulting optimized electronic structure would allow for a detailed and accurate discussion of geometrical CO₂ parameters.

(2) The following comments apply to CO₂-loaded MUF-16(Mn):

- a) Usage of SHEL card should be explicitly mentioned in the SI, and rephrase the following sentence in the SI accordingly "The crystals diffracted to a resolution of just 1.08 Å thus the calculated $\sin(\theta_{\max})/\text{wavelength}$ is 0.463". This should be mentioned in the CIF through VRFs as well.
- b) The ISOR card also makes no sense since the CO₂ was treated isotropically.
- c) Crystal size in CIF 0.1 x 0.1 x 0.2 but in text 0.1 x 0.1 x 0.1, please correct.

(3) All five crystal structures reported need further refinement cycles and/or proper explanations (all A and B level alerts must be removed) in the CIF through VRFs and should be submitted under the same CCDC numbers.

Clarifying comments

We make a few initial comments for clarification. MUF-16 acts as a classical adsorbent. By this we mean that it does not flex or undergo any structural change when it takes up CO₂. This is evident from the SCXRD structures before and after gas adsorption, which are virtually identical. The vast majority of MOF adsorbents act in this classical way, while a smaller subset are flexible and undergo gate opening in response to guest binding. For classical adsorbents, their affinity for particular guests is typically ascribed to a combination of pore size and pore chemistry. In the case of MUF-16, SCXRD provides an atomic level description of the pore size and chemistry before and after CO₂ binding. This provides a detailed picture of the capacity of MUF-16 to act as a host, which is further verified by DFT calculations. Other guests do not bind efficiently, for example methane and the C₂ hydrocarbons. On the other hand, as anticipated N₂O is strongly adsorbed because of its similarity to CO₂ in terms of sizes and electrostatics. Because of the reversed polarity of the hydrocarbon guests with respect to CO₂ they cannot be efficiently adsorbed. In this way we can build up a **precise and detailed mechanistic description of the guest binding process** that is supported by experimental evidence and substantiated by the crystallographic and DFT models. This directly answers the question of why the CO₂ binds efficiently and the other gases bind weakly. While these results permeate the manuscript, we have written in summary of the key points in a paragraph in the conclusion section of the revised manuscript. Now, we and others are equipped to prepare analogues of MUF-16 that present similar binding sites for next-generation adsorption and separation processes.

Reviewer #1 (Remarks to the Author):

This reviewer cannot be convinced by their responses in terms of framework flexibility and gate pressures. Quite a lot parameters will affect their gate pressures including the temperature, pressure, and examined gas molecules. Without clear understanding, the mechanism for this gas separation cannot be rationalized. Current explanation for this gas separation is not reasonable. This work is not ready for the publication yet in NC.

We thank the referee for their comments. Further to our substantial response to their first round of comments, we reiterate that the selectivity of MUF-16 for CO₂ does not rely on a structural change therefore no gate opening is observed in the isotherms. A classical (rather than a flexible) model of is appropriate for MUF-16. This is evidenced by the reported SCXRD structures, which are virtually the same before and after CO₂ adsorption. As outlined above, the mechanism of CO₂ capture is clearly indicated by the SCXRD structure of MUF-16-CO₂ and the DFT calculations, which show how the pore size and electrostatic characteristics of MUF-16 underlie its high affinity. Other guest molecules, such as the C₂ hydrocarbons, are polarised differently to CO₂ therefore are barely taken up by MUF-16 around room temperature. In our estimation this is a clear, reasonable and logical depiction of the gas capture and separation process. We have amended the manuscript to emphasize our description of the mode of action of MUF-16.

Reviewer #2 (Remarks to the Author):

The authors have addressed the questions I raised, and I am ok with the current form.

We thank the referee for their positive comments.

Reviewer #3 (Remarks to the Author):

The authors have addressed most of my questions except the request for water adsorption

experiment. I still think it is important to carry out this experiment especially because the MOF is very hydrophilic. The experiment is fairly simple and not time consuming. I also noted Ref. #48 is missing part of the information.

We thank the referee for their positive comments. We have measured water uptake by MUF-16 and added this isotherm to the manuscript ESI. Ref 48 has also been corrected.

Reviewer #5 (Remarks to the Author):

The manuscript entitled "Selective Capture of Carbon Dioxide from Hydrocarbons Using a Metal-Organic Framework: Relevance to the Purification of Natural Gas and Acetylene" by Omid et al. reports the selective capture of CO₂ from hydrocarbons using MUF-16 (Mn). The manuscript is reviewed with special attention to the crystallographic part as per the editor's request. The overall manuscript is well written, and experiments have been performed to high standards. I recommend this manuscript to be accepted for publication in Nat. Comm. given the authors can address the following crystallographic concerns.

We thank the referee for their positive comments.

(1) The crystallographic work showing the presence of CO₂ inside the MOF pores is convincing but given the lower occupancy (0.77) and large isotropic displacement factors it is not recommended to discuss any CO₂-based geometrical parameters quantitatively. Instead, the authors should employ density functional theory-based calculations to complement their experimental data. This should be a straightforward process using the experimental geometry data as starting point. The resulting optimized electronic structure would allow for a detailed and accurate discussion of geometrical CO₂ parameters.

We have carried out DFT calculations and find a near-perfect agreement with the structure determined by SCXRD. This provides conclusive evidence as to the position and orientation of the CO₂ guest molecule: we now have the weight of both experimental and computational approached which provide identical depictions of the CO₂ binding site in MUF-16.

In light of these additional DFT calculations we have added another author to the manuscript, Dr Ravichandar Babarao.

(2) The following comments apply to CO₂-loaded MUF-16(Mn):

a) Usage of SHEL card should be explicitly mentioned in the SI, and rephrase the following sentence in the SI accordingly "The crystals diffracted to a resolution of just 1.08 Å thus the calculated $\sin(\theta_{\max})/\text{wavelength}$ is 0.463". This should be mentioned in the CIF through VRFs as well.

These updates have been made to the ESI and VRFs have been added to the CIF file.

b) The ISOR card also makes no sense since the CO₂ was treated isotropically.

The only ISOR command used during the refinement of this structure is "ISOR 0.01 0.002 C1". The C1 atom is part of the framework and does not relate to the CO₂ guest molecule.

c) Crystal size in CIF 0.1 x 0.1 x 0.2 but in text 0.1 x 0.1 x 0.1, please correct.

The crystal size has been updated in the text.

(3) All five crystal structures reported need further refinement cycles and/or proper explanations (all A and B level alerts must be removed) in the CIF through VRFs and should be submitted under the same CCDC numbers.

We have revisited the SCXRD structures and eliminated the A and B alerts as far as possible. Where they remain VRF responses have been included in the CIF files as well as the ESI.

Unfortunately, the A and B alerts remain as a consequence of the limitations of the datasets themselves (e.g. the value of $\sin(\theta_{\max})/\lambda$; Poor Data / Parameter Ratio; Low Bond Precision on C-C Bonds). Such alerts are commonly encountered in MOF chemistry due to the inherent characteristics of the MOF crystals. They cannot be eliminated by further refinement cycles and are simply and unavoidable artefact of the CheckCIF process.

REVIEWERS' COMMENTS

Reviewer #5 (Remarks to the Author):

Please see attached pdf.

All (Structural part) looks good expect the following:

When you make VRFs, it is important to split the lines properly in the CIF, otherwise the checkCIF report would be like this (snapshot from the pdf submitted by the authors):

Alert level B
PLAT029 ALERT 3 B _diffn_measured_fraction_theta_full value Low . 0.953 Why?
Author Response: A small number of reflections are absent due to the orientation of the
PLAT084 ALERT 3 B High wR2 Value (i.e. > 0.25) 0.38 Report
Author Response: Despite numerous data collections, the wR2 value remained high due to

If you split your lines in the CIF as follows, the checkCIF would look good. So please revise the checkCIFs as seen below:

```
;  
PROBLEM: _diffn_measured_fraction_theta_full value Low . 0.953 Why?  
RESPONSE: A small number of reflections are absent due to the  
orientation of the crystal with respect to the diffractometer.  
;  
_vrf_PLAT084_muf-16-assynthesized  
;  
PROBLEM: High wR2 Value (i.e. > 0.25) ..... 0.38 Report  
RESPONSE: Despite numerous data collections, the wR2 value  
remained high due to an inherent lack of precise ordering in the material.  
;
```

Alert level B
PLAT029_ALERT_3_B _diffn_measured_fraction_theta_full value Low . 0.953 Why?
Author Response: A small number of reflections are absent due to the orientation of the crystal with respect to the diffractometer.
PLAT084_ALERT_3_B High wR2 Value (i.e. > 0.25) 0.38 Report
Author Response: Despite numerous data collections, the wR2 value remained high due to an inherent lack of precise ordering in the material.

Reviewer #5:

We have changed the cif files, as requested.